# Differential Effects of Two Isocaloric Healthy Diets on Postprandial Lipid Responses in Individuals with Type 2 Diabetes

**DOI:** 10.3390/nu16030333

**Published:** 2024-01-23

**Authors:** Giuseppina Costabile, Dominic Salamone, Giuseppe Della Pepa, Marilena Vitale, Roberta Testa, Paola Cipriano, Giuseppe Scidà, Angela Albarosa Rivellese, Giovanni Annuzzi, Lutgarda Bozzetto

**Affiliations:** 1Department of Clinical Medicine and Surgery, Federico II University, Via Sergio Pansini 5, 80131 Naples, Italy; giuseppina.costabile@unina.it (G.C.); dominic.salamone@unina.it (D.S.); marilena.vitale@unina.it (M.V.); roberta.testa@unina.it (R.T.); paolacipriano2006@libero.it (P.C.); giuseppe.scida@unina.it (G.S.); rivelles@unina.it (A.A.R.); annuzzi@unina.it (G.A.); lutgarda.bozzetto@unina.it (L.B.); 2Cardiometabolic Risk Unit, Institute of Clinical Physiology, National Research Council-CNR, Via Giuseppe Moruzzi 1, 56124 Pisa, Italy

**Keywords:** postprandial triglycerides, dietary interventions, monosaturated fatty acids, polyphenols, type 2 diabetes

## Abstract

Background. High blood concentrations of triglycerides (TG) in the postprandial period have been shown to be more closely associated with the risk of cardiovascular disease (CVD) than fasting values in individuals with type 2 diabetes (T2D). Dietary changes are the primary determinants of postprandial lipid responses. Methods. We investigated the effects of an isocaloric multifactorial diet, rich in n-3 PUFA, MUFA, fiber, polyphenols, and vitamins, compared to an isocaloric diet, containing the same amount of MUFA, on the postprandial lipid response in T2D individuals. Following a randomized, controlled, parallel group design, 43 (25 male/18 female) T2D individuals were assigned to an isocaloric multifactorial (n = 21) or a MUFA-rich diet (n = 22). At the beginning and after the 8 weeks of dietary intervention, the concentrations of plasma triglycerides, total cholesterol, HDL cholesterol, and non-HDL cholesterol were detected at fasting and over a 4-h test meal with the same composition as the prescribed diet. Results. The concentrations of fasting plasma triglycerides, total cholesterol, HDL cholesterol, and non-HDL cholesterol did not change after both diets. Compared with the MUFA diet, the 8-week multifactorial diet significantly lowered the postprandial response, which was evaluated as the incremental area under the curve (iAUC), of triglycerides by 33% (64 ± 68 vs. 96 ± 50 mmol/L·240 min, mean ± SD, respectively, *p* = 0.018), total cholesterol by 105% (−51 ± 33 vs. −25 ± 29, *p* = 0.013), and non-HDL cholesterol by 206% (−39 ± 33 vs. −13 ± 23, *p* = 0.013). Conclusions. In T2D individuals, a multifactorial diet, characterized by several beneficial components, improved the postprandial lipid response compared to a MUFA diet, generally considered a healthy diet being reduced in saturated fat, and probably contributed to the reduction of cardiovascular risk.

## 1. Introduction

Postprandial dyslipidemia, particularly abnormal concentrations of triglycerides (TG), has been shown to be more closely associated with cardiovascular disease (CVD) risk, than fasting values [1,2]. Moreover, in type 2 diabetes (T2D), different abnormalities characterize postprandial lipid metabolism—i.e., increased concentration of circulating triglycerides-rich lipoproteins and atherogenic particles of LDL and HDL—that could contribute to very high CV risk [3,4,5,6,7]. Diet is an important component of lifestyle, which due to its pleiotropic action, represents a natural approach to modulate postprandial changes as well other CVD risk factors [8,9]. Evidence from clinical trials shows that diets rich in whole grains and fiber are able to reduce plasma lipid concentrations in the postprandial state in individuals with T2D or metabolic syndrome [8,9,10,11]. As for dietary fats, the replacement of dietary sources of saturated fatty acids (SFAs) with sources of polyunsaturated fatty acids (PUFAs) or monounsaturated fatty acids (MUFAs) does not induce significant variations in the postprandial lipid response in healthy adults, while instead it seems to be most effective in individuals at high cardiometabolic risk [12,13,14]. Among the PUFAs, PUFA n-3 seems to have the most favorable effects on the postprandial triglyceride response, although this is in general with high dosages [15,16,17]. Among other components of the diet, polyphenols have also been shown to act on postprandial lipid response modulation [18]; in fact, the results of an 8-week nutritional intervention trial showed that a diet with a natural high content of polyphenols (3 g/day) significantly reduced plasma triglyceride concentrations at fasting and postprandially in overweight/obese individuals with metabolic syndrome [19].

Based on this background, since there is no evidence from studies evaluating the combined effect of these aforementioned dietary components on postprandial lipid response, the aim of the current clinical trial is to evaluate the effect of a multifactorial diet, rich in various beneficial dietary components (i.e., n-3 PUFA, MUFA, fiber, polyphenols, and vitamins), compared to an isocaloric diet, containing the same amount of MUFA and considered to be a healthy dietary approach being reduced in saturated fat, on lipid response in the postprandial state in individuals with T2D. This study is an exploratory analysis of a clinical trial in which the principal endpoint was the reduction in liver fat content [20].

## 2. Materials and Methods

### 2.1. Participants

The details regarding the study design, characteristics of the participants, and dietary interventions have been described in detail previously [20]. In brief, the study participants were individuals with T2D and abdominal obesity of both genders, aged 35–75 years, with the following criteria of inclusion: a high waist circumference (>88 cm for women and >102 cm for men); optimal glucose control with glycated hemoglobin (HbA1c) concentrations ≤7.5%; low-density lipoprotein (LDL) cholesterol <3.37 mmol/L; and fasting plasma concentrations of triglyceride <3.96 mmol/L with or without a stable treatment with lipid-lowering drugs, which if taken by the patients at the screening visit, were continued unchanged throughout the study. With regard to T2D treatment, apart from diet, only stable treatment with oral glucose-lowering drugs (repaglinide, sulfonylureas, metformin, and dipeptidyl peptidase-4 inhibitors) was allowed. The exclusion criteria were represented by acute/chronic diseases severely affecting health status, current smoking, regular moderate/strenuous physical activity, prescription of antioxidants/vitamins/nutraceuticals, and changes (±3 kg) in body weight over the last six months. 

The study protocol was approved by the Ethics Committee of Federico II University and carried out in accordance with the Declaration of Helsinki for clinical trials. Written informed consent was provided by all participants. The study was registered at ClinicalTrials.gov (NCT03380416).

### 2.2. Study Design

This study included a randomized, controlled, parallel-group trial, and participants were assigned to a MUFA or multifactorial diet for 8 weeks [21,22]. The two diets were isoenergetic, to avoid body weight changes over the course of the study, and provided the same content in total carbohydrates, fat, MUFA, and protein while differing for fiber, polyphenols, glycemic index, n-3 and n-6 PUFAs, and vitamins [20]. 

To investigate the postprandial lipid response at baseline and after the 8 weeks of dietary intervention, after 12 h of overnight fasting, the participants consumed the same test meal with a similar composition as the diet assigned by the randomization (Table 1).

The test meal consumed by participants assigned to the MUFA diet was composed of rice, peas, tomato sauce, extra-virgin olive oil, fillet steak, beef cured meat (bresaola), egg, and banana [Energy 843 kcal, protein 18% total energy (TE), fat 42% TE, carbohydrate 40% TE]. The test meal consumed by participants assigned to the multifactorial diet was composed of beans, pasta, extra-virgin olive oil, rucola, salmon, decaffeinated green tea, and orange [Energy 821 kcal, protein 18% total energy (TE), fat 42% TE, carbohydrate 40% TE].

Metabolic parameters were evaluated in the fasting state and postprandially (after the test meal) before and after the 8 weeks of the intervention period. A 7-day food record completed by the participants at 4 and 8 weeks was collected for the evaluation of dietary adherence. The participants were given a weekly dietary plan to warrant that the amount and frequency of each food that characterized the two experimental diets were satisfied. Furthermore, a skilled dietitian saw participants every 2 weeks to further improve adherence to dietary intervention. Finally, for the whole duration of the study, some foods were given to participants to facilitate dietary compliance.

### 2.3. Laboratory Methods

Anthropometrics and metabolic data were evaluated at the beginning and after the 8-week dietary intervention period. Blood samples were collected at fasting and 60, 120, 180, and 240 min after the test meal to measure plasma triglycerides, total cholesterol, and HDL cholesterol. Plasma glucose and lipids [total cholesterol, high-density lipoprotein (HDL) cholesterol, and triglyceride concentrations] were detected via enzymatic colorimetric methods (Roche Diagnostics, Milan, Italy, and ABX Diagnostics, Montpellier, France) using an ABX Pentra 400 (HORIBA Medical, Montpellier, France). HbA1c was measured via HPLC (Agilent HPLC 1200, Santa Clara, CA, USA). Plasma insulin concentrations were measured via ELISA (DIA-source ImmunoAssay S.A., Nivelles, Belgium) using a Triturus Analyzer (Diagnostic Grifols, S.A., Barcelona, Spain). The LDL cholesterol (Friedewald formula [23]) and non-HDL cholesterol (total cholesterol-HDL cholesterol values) concentrations were calculated. The homeostatic model assessment of insulin resistance (HOMA-IR) was calculated using the following formula: fasting glucose (mmol/L)·fasting insulin (pmol/L)/22.

### 2.4. Statistical Analysis

The sample size (23 participants for each group) was calculated based on the primary outcome of the study—i.e., liver fat content percentage reduction [20]. However, to evaluate the 8-week effect of the multifactorial diet on postprandial lipids, this sample was also adequate for the current exploratory analysis. In fact, based on our previous trial, in which we observed an 18% reduction in triglyceride iAUC in patients with T2D following a plant-based high-carbohydrate/high-fiber diet versus a high-MUFA/low-carbohydrate diet [10], 48 participants (24 for each group) had to be studied to observe a 30% difference in postprandial triglyceride responses between the two dietary interventions with a 0.05 significance level and 80% power (type II error = 0.02), considering a 15% drop out rate.

Data are expressed as mean ± standard deviation (M ± SD) unless otherwise stated. Plasma triglycerides, total cholesterol, HDL cholesterol, and non-HDL cholesterol are reported as increased/decreased from fasting values, calculated by subtracting the fasting value from that of each time point in the group. The overall postprandial lipid responses were assessed using a general linear model (GLM) repeated measures analysis during the entire postprandial response (240 min) to evaluate the effects of time, meal, and time–meal interactions. In detail, the lipid concentrations at time 60, 120 and 240 min were added as levels of the within-subject “time” factor, and the multifactorial meal and MUFA meal were added as levels of the within-subject “meal”. The trapezoidal rule was used to calculate the incremental/decremental area under the curve above the fasting value (iAUC). Within-group differences (8-week vs. baseline values) were evaluated using the Wilcoxon signed rank non-parametric test for dependent samples. At baseline and at the end of the intervention, between-group differences were evaluated via the Wilcoxon–Mann–Whitney non-parametric test for independent samples. Furthermore, the differences found at 8 weeks of the intervention were adjusted for baseline values when appropriate (ANCOVA). The categorical variables were evaluated using the χ^2^ test. The bivariate associations were assessed via a Spearman correlation. All analyses were performed according to the intention-to-treat principle. A *p* value < 0.05 was considered significant. The statistical analysis was performed using the SPSS software 27.0 (SPSS/PC; IBM, Armonk, NY, USA).

## 3. Results

### 3.1. Anthropometrics and Fasting Measurements

In total, 49 individuals were randomly assigned to the two experimental dietary interventions, and 43 (MUFA diet n = 22, multifactorial diet n = 21) completed the trial (Appendix A). The adherence to dietary interventions was optimal [20].

Participants assigned to the two dietary interventions were similar in terms of anthropometric and biochemical parameters at the baseline. Participants were also comparable regarding glucose- and lipid-lowering drugs, which did not change during the study duration. At the end of 8 weeks of the dietary interventions, a slight and similar significant reduction in body weight was observed for both diets, without significant differences between the groups (Table 2). Similarly, a significant reduction in HbA1c levels was observed after both dietary interventions, without a statistically significant difference between the groups. Fasting plasma lipids, glucose, insulin, and HOMA-IR did not change in both groups after the dietary interventions (Table 2).

### 3.2. Postprandial Lipid Response

At baseline, before the dietary intervention, the postprandial triglyceride profile following the meal test was lower in the multifactorial group compared with the MUFA meal group (*p* = 0.015, time·meal), showing a significant difference among groups particularly at 120 min after the meal (*p* = 0.020) (Figure 1, Panel A), which highlights the acute effect of the multifactorial meal. At the end of the 8 weeks of dietary intervention, the postprandial triglyceride profile remained lower in the multifactorial group compared with the MUFA group (*p* = 0.020 time·meal), showing a significant difference among groups at 60, 120 and 240 min after the meal (*p* < 0.05) (Figure 1, Panel A).

The postprandial triglyceride response, evaluated as iAUC up to 240 min, did not differ significantly among groups at baseline (86 ± 50 vs. 65 ± 56 mmol/L·240 min, MUFA vs. multifactorial, respectively; *p* = 0.416), (Figure 1, Panel B). After 8 weeks, it was significantly decreased in the multifactorial group (−33%) compared with the MUFA group (64 ± 68 vs. 96 ± 50 mmol/L·240 min, respectively; *p* = 0.032) (Figure 1, Panel B).

The postprandial total cholesterol profile decreased, but the reduction was more evident in the multifactorial meal group compared with the MUFA meal group, both at baseline—i.e., before the dietary intervention (*p* = 0.039, time·meal) and after 8 weeks (*p* = 0.001 time·meal), (Figure 1, Panel C). In particular, after 8 weeks, a significant difference between the groups at 60, 120 and 240 min after the meal was found (*p* < 0.05) (Figure 1, Panel C). The postprandial total cholesterol, iAUC, did not change significantly in both groups before the dietary intervention (−39 ± 34 vs. −48 ± 37 mmol/L·240 min, MUFA vs. multifactorial, respectively; *p* = 0.296) at baseline; after 8 weeks, it lowered significantly more in the multifactorial group (−105%) compared with the MUFA group (−51 ± 33 vs. −25 ± 29 mmol/L·240 min, *p* = 0.015) (Figure 1, Panel D).

The HDL cholesterol profile also decreased in the postprandial period. This decrease was more evident in the multifactorial meal group at baseline (*p* = 0.025 time·meal), while the difference only tended to reach statistical significance after 8 weeks (*p* = 0.074 time·meal) (Figure 2, Panel A).

The postprandial HDL cholesterol, iAUC, decreased similarly in the two groups at baseline (−19 ± 12 vs. −18 ± 10 mmol/L·240 min, MUFA vs. multifactorial, respectively; *p* = 0.993) and after 8 weeks (−17 ± 10 vs. −11 ± 11 mmol/L·240 min, *p* = 0.212) (Figure 2, Panel B).

Before the dietary intervention, the non-HDL cholesterol postprandial profile was similar among the groups (*p* = 0.084 time·meal) (Figure 2, Panel C). After 8 weeks, it decreased more in the multifactorial group compared with the MUFA group (*p* = 0.001 time·meal) with a significant difference between groups at 60,120 and 240 min after the meal (*p* < 0.05), (Figure 2, Panel C). Similar results were obtained when evaluating the postprandial non-HDL cholesterol, iAUC, that did not change between the two groups at baseline (−20 ± 26 vs. −30 ± 28 mmol/L·240min, MUFA vs. multifactorial, respectively; *p* = 0.126) (Figure 2, Panel D), while it decreased more in the multifactorial group (−206%) than in the MUFA (−39 ± 33 vs. −13 ± 23 mmol/L·240 min, *p* = 0.002) (Figure 2, Panel D).

To untangle the impact of the acute meal effect and the 8-week effect of dietary interventions on postprandial lipid changes between the groups, the differences found after 8 weeks of dietary intervention were adjusted for the values of the baseline. The differences between the groups remained significant with a greater effect in terms of percentage reduction particularly for triglycerides (triglycerides: −113%, *p* = 0.045; total cholesterol: −116%, *p* = 0.022; and non-HDL cholesterol: −223%; *p* = 0.005), when comparing the multifactorial diet with the MUFA diet. 

### 3.3. Postprandial Glucose and Insulin Response

The postprandial glucose profile following the meal test did not change in the multifactorial group compared with the MUFA meal group at baseline (*p* = 0.095, time·meal) and after 8 weeks of dietary intervention (*p* = 0.065, time·meal) (Appendix A, Panel A). Similarly, the postprandial glucose response, evaluated as iAUC up to 240 min after the meal, did not change between the two groups at baseline (508 ± 335 vs. 467 ± 374 mmol/L·240 min, MUFA vs. multifactorial, respectively; *p* = 0.717) and after 8 weeks of dietary intervention (462 ± 333 vs. 508 ± 397 mmol/L·240 min, MUFA vs. multifactorial, respectively; *p* = 0.719), (Appendix A, Panel B).

The postprandial insulin profile following the meal test did not change in the multifactorial group compared with the MUFA group at baseline (*p* = 0.063, time·meal) (Appendix A, Panel C). After 8 weeks, it increased more in the multifactorial group compared with the MUFA group, particularly at 30 and 60 min after the meal (*p* < 0.05), with an overall significant difference between the groups (*p* = 0.004 time·meal), (Appendix A, Panel C). Similarly, the postprandial insulin response, evaluated as iAUC up to 240 min after the meal, did not differ between the two groups at baseline (61,618 ± 32,819 vs. 54,111 ± 39,889 pmol/L·240 min, MUFA vs. multifactorial, respectively; *p* = 0.508), while it significantly increased after 8 weeks of dietary interventions in the multifactorial group as compared to the MUFA group (56,250 ± 22,778 vs. 65,479 ± 36,694 pmol/L·240 min, MUFA vs. multifactorial, respectively; *p* = 0.011) (Appendix A, Panel D). 

## 4. Discussion

In the current study, we show that consuming a multifactorial diet, naturally rich in dietary fibers, polyphenols, MUFA, PUFA, PUFA n-3 particularly, and other bioactive components for 8 weeks improves the postprandial lipid response in T2D patients. In particular, this type of diet, compared to a MUFA diet, is effective in significantly reducing the plasma concentrations of triglycerides and cholesterol in the postprandial state by approximately 33% and 105%, respectively, and non-HDL cholesterol levels by 206%. These effects were also acutely present at baseline before the dietary intervention and after a single meal resembling the multifactorial diet, but they were more consistent after 2 months on the same diet. Instead, the observed decrease in HDL cholesterol was somewhat more evident in acute conditions after a single meal similar to the multifactorial diet than after 2 months.

The reduction in postprandial triglycerides reported in the present trial is relevant from a clinical perspective, considering that hypertriglyceridemia in the postprandial state is a well-known independent risk factor for CVD, and the magnitude of the observed reduction might lead to a significant decrease in CVD risk incidence in our patients [1,2]. 

In patients with T2D, a similar reduction in the postprandial triglyceride response was observed after a 4-week nutritional intervention with an isoenergetic diet rich in carbohydrates and fiber, in comparison with a diet low in carbohydrates and high in MUFA, while no variations were reported in the postprandial plasma cholesterol [9]. In line with the present findings, a diet rich in whole-grain products decreased the postprandial triglyceride response by 43% in overweight/obese patients with metabolic syndrome after 12 weeks of treatment [9]. Similarly, the consumption of a diet composed of foods naturally rich in polyphenols for 8 weeks significantly reduced the fasting and postprandial plasma triglyceride concentrations in high cardiometabolic risk individuals [19]. 

Considering the various features of the multifactorial diet—i.e., whole grain, dietary fiber, and polyphenols—inspired by the traditional Mediterranean diet and sharing components of other healthy dietary patterns, such as the Nordic diet and the prudent diet, it is likely that its efficacy on postprandial lipid reduction might depend on the synergic effect of the different dietary components [24,25,26].

Polyphenols and dietary fiber might inhibit the activity of pancreatic lipase, favoring a reduction in the intestinal hydrolysis of triglycerides, blunting their absorption, and reducing the synthesis of chylomicrons [10,27,28]. Dietary fiber might slow down and reduce cholesterol absorption in the intestine as well as increase the amount of fecal bile acids and the synthesis of bile acids from cholesterol in the hepatocytes [29,30,31,32]. Acting on intestinal triglycerides and cholesterol absorption, dietary fiber and polyphenols might also reduce substrate availability for the production of very low-density lipoprotein (VLDL) in the hepatocytes [10], which are generally increased in the setting of insulin resistance and T2D, and contribute to postprandial dyslipidemia [1,2]. In this regard, the fact that dietary components might directly favor a reduction of de novo-lipogenesis and VLDL synthesis, as reported in our previous study [21], should not be excluded. It is of note that dietary components might also modulate the clearance of chylomicrons and VLDL by acting on lipoprotein lipases and their cofactors on the surface of vascular endothelial [10,29,32].

Another important observation coming from our data is that the decrease in postprandial total cholesterol with the multifactorial diet is not only related to the major reduction in HDL cholesterol but is perfectly in line with the greatest decrease obtained with this diet in non-HDL cholesterol. This means that in the postprandial period, the multifactorial diet may induce a more significant reduction in lipoproteins of intestinal origin, in particular chylomicrons and remnants of chylomicrons, which are considered highly atherogenic. The greatest decrease in non-HDL cholesterol even in the postprandial period and therefore, for a long period of time during the day, may be clinically relevant for T2D individuals considering that non-HDL cholesterol concentrations are strongly linked with CVD risk in the long term [33,34,35]. Notably, in our study, the observed great values in postprandial triglyceride and non-HDL cholesterol reduction occurs in individuals with T2D in optimal glucometabolic control and on top of glucose- and lipid-lowering drugs; these results highlight the relevant additive role of the diet, particularly the combination of different dietary components characterizing the traditional Mediterranean diet, on the postprandial lipid profile.

It is worthy of note that the observed results were reached in patients who did not habitually perform moderate or strenuous physical activity. Evidence consistently supports the concept that physical activity beneficially impacts postprandial lipemia, and a meta-analysis shows that physical activity reduced the iAUC of triglycerides by 23% [36]. In our trial, the absence of regular physical activity in the trial’s inclusion criteria allowed us to further highlight the unique role of diet quality and its different nutrient content on postprandial lipemia.

The multifactorial diet also induces a greater decrease in HDL cholesterol, compared to the MUFA diet, but this effect is of borderline significance, particularly after 8 weeks of the intervention. 

Previous studies observed a reduction in fasting HDL cholesterol levels after the consumption of diets rich in polyphenols and dietary fiber [19,37,38,39,40], and this finding, at first glance, could be considered an adverse effect in terms of CVD risk. However, evidence shows that the beneficial effects of HDL in decreasing CVD risk are not fully reflected through the amount of cholesterol carried by the HDL; rather, the quality of the HDL particles, i.e., their functions in the cholesterol efflux capacity and their beneficial properties in oxidative stress, inflammation, and the endothelium [40], may be more relevant. In our study, we cannot exclude the fact that despite the observed reduction in HDL cholesterol, the multifactorial diet could have improved the quality composition and function of HDL cholesterol as reported with a similar dietary pattern [40,41]. The impact of different dietary components on HDL cholesterol concentrations and functional characteristics in the postprandial state and its role in the modulation of CVD risk is poorly understood, indicating a need for additional research. 

Postprandial increases in glucose concentration and insulin resistance, or compensatory hyperinsulinemia, are widely known independent CVD risk factors and have been suggested as major determinants of postprandial lipemia [42,43]. 

In our study, we can exclude the possible impact of a postprandial glucose response on the postprandial lipid response, considering the two dietary interventions did not differently modify the glucose profile after the meal; furthermore, no correlations have been found between the postprandial (iAUC_0–240_) triglyceride and glucose responses at baseline (r = −0.117, *p* = 0.461) and after 8 weeks (r = −0.185, *p* = 0.242) of dietary interventions.

Fasting insulin resistance was not affected by the dietary interventions, probably due to the not excessively high values of HOMA-IR and the stable treatment with glucose-lowering drugs affecting insulin sensitivity, such as metformin. Interestingly, the multifactorial diet induced a significant increase in the early postprandial insulin response, which is considered a marker of β-cell functionality. In fact, consistent data supports the view that an early β-cell functional defect is a major contributor to the failure of long-term metabolic control in T2D, and also represents a mechanism underlying the pathogenesis of hyperglycemia in the majority of individuals who develop T2D [44]. 

The increase in the early postprandial insulin response could be advocated as a possible mechanism involved in the postprandial triglyceride reduction observed in the multifactorial diet. Insulin exerts an antilipolytic effect through hormone-sensitive lipase, promotes triglyceride synthesis, and activates lipoprotein lipase in adipose tissue, which is responsible for the clearance of triglycerides from plasma [9]. However, in our study, no correlation was found between postprandial triglyceride and insulin responses at baseline (r = 0.154, *p* = 0.330) and after 8 weeks (r = 0.056, *p* = 0.726) of dietary intervention.

Our study presents several strengths such as the randomized and controlled design as well as the optimal adherence to the two experimental diets, which were also confirmed by the variations in the serum triglyceride fatty acid profile—i.e., the PUFAs significantly increased in the triglycerides only after the multifactorial diet, whereas a significant increase in oleic acid (the major source of the MUFA characterizing the two diets) was observed after both experimental dietary interventions [21]. Some limitations should also be considered. First, only patients with T2D in optimal blood glucose control and with fasting plasma lipid levels in the normal range were included in the study. Consequently, our results cannot be extrapolated to all T2D patients. The small sample size represents a further limitation of the study. Again, a dietary intervention of 8 weeks cannot represent a real long-term experimental intervention, although it is probably adequate to act on lipid profile changes. Finally, a longer (>4 h) evaluation of lipid responses measured after a meal should provide a good estimation of postprandial lipemia [45,46]. However, evidence shows that a reduced 4-h postprandial lipemia test is an accurate and replicable index for an 8-h test [47], and the postprandial triglyceride concentrations detected 2–4 h after a meal might also be greatly related to CVD risk [48].

## 5. Conclusions

The findings of our study show that a multifactorial diet—rich in fibers, polyphenols, PUFA, mainly n-3, MUFA, and other bioactive compounds—improves postprandial lipid response in T2D patients compared to a MUFA diet, generally considered a healthy diet being reduced in saturated fat. Therefore, the multifactorial diet might contribute to reducing cardiovascular risk in individuals with T2D.

Further clinical trials with a larger sample size and longer duration, also considering patients with T2D with severe lipid abnormalities, are required to corroborate the favorable effects of a multifactorial diet on lipid response in the postprandial state.

## Figures and Tables

**Figure 1 nutrients-16-00333-f001:**
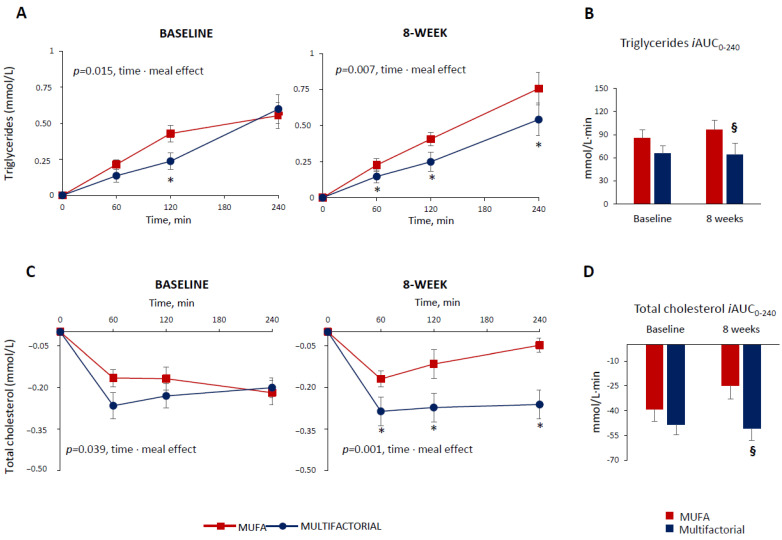
Plasma triglycerides and total cholesterol responses to a test meal evaluated as increased/decreased from fasting values (Panel (**A**,**C**)) and iAUC (Panel (**B**,**D**)) at baseline and after 8 weeks of dietary intervention with the MUFA (red lines and bars) (n = 22) or multifactorial diet (blue lines and bars) (n = 21). Data are mean ± SEM. * *p* < 0.05 Multifactorial vs. MUFA diet, unpaired *t*-test; § *p* < 0.05 Multifactorial vs. MUFA, 8-week values are adjusted for baseline via ANCOVA.

**Figure 2 nutrients-16-00333-f002:**
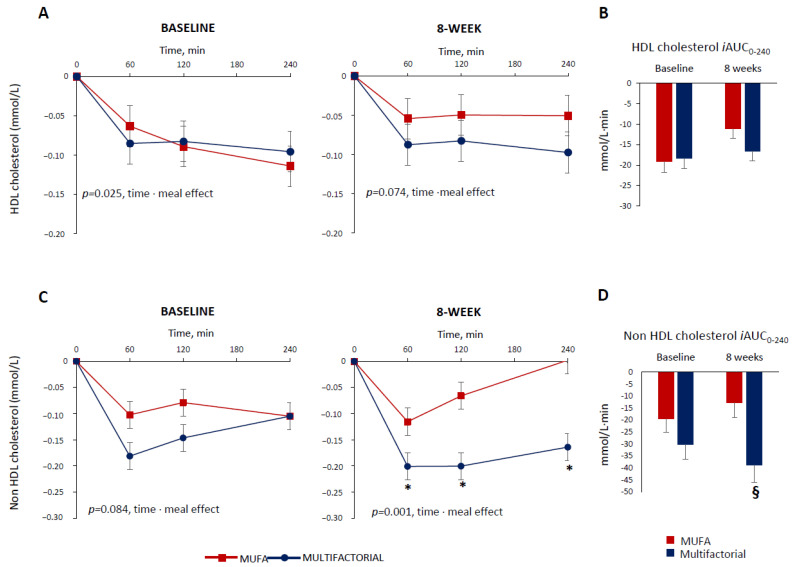
Plasma HDL cholesterol and non-HDL cholesterol responses to a test meal evaluated as increased/decreased from fasting values (Panel (**A**,**C**)) and iAUC (Panel (**B**,**D**) )at baseline and after 8 weeks of dietary intervention with the MUFA (red lines and white bars) (n = 22) or multifactorial diet (blue lines and bars) (n = 21). Data are mean ± SEM. * *p* < 0.05 Multifactorial vs. MUFA diet, unpaired *t*-test; § *p* < 0.05 Multifactorial vs. MUFA, 8-week values are adjusted for baseline via ANCOVA.

**Table 1 nutrients-16-00333-t001:** Composition of the two test meals.

	MUFA Diet	Multifactorial Diet
Energy (kcal)	843	821
Protein (% TEI)	18	18
Fat (% TEI)	42	42
Saturated (% TEI)	7	7
Monounsaturated (% TEI)	28	28
Polyunsaturated (% TEI)	3.4	4.5
n-3 (g)	0.36	1.20
n-6 (g)	2.87	2.91
Carbohydrates (% TEI)	40	40
Sugars (% TEI)	7.6	7.9
Fiber (g/1000 kcal)	7.7	14.2
Glycemic index (%)	84	39
Glycemic load	70	32
Vitamin E (mg)	8.3	8.5
Vitamin D (μg)	1.83	2.72
Vitamin C (mg)	33	168
Polyphenols (mg)	16	758
Total ORAC (μmolT)	1584	4441

TEI: total energy intake; ORAC: Oxygen radical absorbance capacity.

**Table 2 nutrients-16-00333-t002:** Anthropometric, clinical, and fasting metabolic parameters of participants at baseline and after 8 weeks of dietary interventions.

	MUFA Diet (n = 22)		Multifactorial Diet (n = 21)		
	Baseline	8 Weeks	Baseline	8 Weeks	*p* ^§^
Gender (Male/Female)	12/10	-	13/8	-	-
Age (years)	63 (5)	-	63 (7)	-	0.784
Body weight (kg)	84 (14)	83 (14) *	85 (12)	84 (12) *	0.951
BMI (Kg/m^2^)	31(3)	30(3) *	32(4)	31(4)	0.894
Waist circumference (cm)	104 (11)	104 (11)	106 (10)	106 (10)	0.495
Hb1Ac (%)	6.6 (0.6)	6.4 (0.7) *	6.6 (0.5)	6.3 (0.5)	0.760
Glucose (mmol/L)	7.2 (0.9)	7.1 (0.9)	6.8 (0.8)	7.0 (1.0)	0.480
Insulin (pmol/L)	126 (68)	133 (79)	130 (72)	110 (56)	0.120
HOMA-IR	5.8 (3.0)	6.1 (3.6)	5.6 (2.7)	4.9 (2.6)	0.256
Total cholesterol (mmol/L)	3.70 (0.7)	3.57 (0.7)	3.85 (0.8)	3.83 (0.9)	0.372
HDL cholesterol (mmol/L)	1.01 (0.2)	0.98 (0.2)	1.09 (0.3)	1.01 (0.2)	0.115
Plasma triglycerides (mmol/L)	1.21 (0.4)	1.18 (0.5)	1.29 (0.3)	1.58 (0.6)	0.496
LDL cholesterol (mmol/L)	2.12 (0.6)	2.02 (0.5)	2.18 (0.5)	2.23 (0.6)	0.192
Non-HDL cholesterol (mmol/L)	2.69 (0.1)	2.56 (0.1)	2.77 (0.1)	2.82 (0.2)	0.146
Lipid-lowering drugs	14 (58%)	-	10 (42%)	-	0.364
Glucose-lowering drugs	17 (55%)	-	14 (45%)	-	0.438

Mean (SD). * *p* < 0.05 vs. baseline (paired sample *t*-test or χ^2^). ^§^ Comparison between groups that were made using 8-week values were adjusted for baseline values. BMI, body mass index; HbA1c, glycated hemoglobin; HOMA-IR, homeostatic model assessment of insulin resistance; HDL, high-density lipoprotein; LDL, low-density lipoprotein.

## Data Availability

Data are available on reasonable request by contacting the corresponding author.

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
