# Peer review of "Differential Effects of Two Isocaloric Healthy Diets on Postprandial Lipid Responses in Individuals with Type 2 Diabetes"

_nutrients, 2024, doi:10.3390/nu16030333_

Round 1
Reviewer 1 Report
Comments and Suggestions for Authors
The main question addressed by the research: are there differences in postprandial lipid responses after two different healthy diets. The topic is original, because responses and actual effect of a certain diet in a human lack high quality studies. Therefore this study addresses a specific gap in the field.
This publication adds the information regarding postprandial response. I do not see any need for a specific improvement. To increase the quality of this study only increasing the number of participants would give higher impact.
The conclusions are consistent with the evidence and arguments presented. They address the main question posed.
The references are appropriate.
Author Response
Response to the comments by the Reviewer #1
The main question addressed by the research are there differences in postprandial lipid responses after two different healthy diets. The topic is original, because responses and actual effect of a certain diet in a human lack high quality studies. Therefore, this study addresses a specific gap in the field.
This publication adds the information regarding postprandial response. I do not see any need for a specific improvement. To increase the quality of this study only increasing the number of participants would give higher impact.
The conclusions are consistent with the evidence and arguments presented. They address the main question posed.
The references are appropriate.
Response: We thank the reviewer for their time and consideration of our manuscript.
We agree with the reviewer that a larger sample size could had improve the quality of the study as reported in the Conclusions “Further clinical trials, with a larger sample size and longer duration...” See line 339.
In the revised manuscript, this point has been also highlighted as a limitation of the study.
“The small sample size represents a further limitation of the study.” See lines 326-327.

Reviewer 2 Report
Comments and Suggestions for Authors
1. The authors should explain how they arrived at the sample size.
2. Instead of log transformation, the authors could have used non-parametric testing.
3. Did analyses follow the intention-to-treat principle?
4. The authors should present the overall change (AUC for a given variable over time) and only then present the time-specific differences. If the overall curve is not significant, there is no reason to present the time-specific differences.
5. It is not clear to me that the difference in post prandial lipid patterns reflects the eight weeks prior to the meal test. The differences could just as easily represent acute differences in meal composition.
6. What evidence of compliance with dietary intervention do the authors have?
7. Lowering HDL with a multifactorial diet seems like an unintended adverse event.
8. The authors might want to consider changing the name of “multifactorial diet” to something like “prudent diet” or some other title more descriptive than the title they’ve chosen.
Author Response
Response to the comments by the Reviewer #2
We greatly appreciate the Reviewer’s time spent evaluating our study. We have considered his/her suggestions and have accordingly rewritten the manuscript. The changes in the new version of the text are written in red.
1. The authors should explain how they arrived at the sample size.
Response: as reported in the Statistical analysis, the sample size was calculated based on the primary outcome of the study, i.e. liver fat content percentage reduction, for which 23 patients for each dietary intervention were needed to study. However, this sample was also adequate for the current exploratory analysis.
In fact, in a previous our trial (see reference 10, Bozzetto, L. et al. Acta Diabetol 2014, 51, 385-93), we had observed a 18% reduction in triglyceride iAUC in patients with type 2 diabetes following a plant-based high-carbohydrate/high-fiber diet versus a high-MUFA/low-carbohydrate diet. In the current trial, we have hypothesized that a Multifactioral diet (rich not only in fiber but also in other components) might further induce a 30% difference in triglyceride iAUC compared with a high-MUFA diet. To this aim, forty-eight participants (24 for arm) had to be studied to observe a 30% difference in postprandial triglyceride response between the two dietary interventions with 0.05 significance level and 80% power (type II error=0.02), considering a 15% drop out rate.
In the revised manuscript we have clarified this point.
“In fact, based on our previous trial in which we observed a 18% reduction in triglyceride iAUC in patients with T2D following a plant-based high-carbohydrate/high-fiber diet versus a high-MUFA/low-carbohydrate diet [10], forty-eight participants (24 for each group) had to be studied to observe a 30% difference in postprandial triglyceride response between the two dietary interventions with 0.05 significance level and 80% power (type II error=0.02), considering a 15% drop out rate.” See lines 132-138.
2. Instead of log transformation, the authors could have used non-parametric testing.
Response: As suggested, we have performed statistical analyses with non-parametric tests in the revised manuscript.
“Within-group differences (8-week vs. baseline values) were evaluated by Wilcoxon signed rank non-parametric test for dependent samples. At baseline and at the end of the intervention, between-group differences were evaluated by Wilcoxon-Mann-Whitney non-parametric test for independent samples.” See the Statistical Analysis paragraphs, lines 149-155.
The new analyses confirmed the differences observed with the previous parametric tests; in the revised manuscript, the p values have been updated in the Results section according to the non-parametric test.
3. Did analyses follow the intention-to-treat principle?
Response: we thank the reviewer for this points that has been clarified in the revised manuscript.
“All analyses were performed according to the intention-to-treat principle.” See lines 154-155.
4. The authors should present the overall change (AUC for a given variable over time) and only then present the time-specific differences. If the overall curve is not significant, there is no reason to present the time-specific differences.
Response: We thank the reviewer for this feedback. We have removed the significance at 240 min from the postprandial HDL response graph (Figure 2, Panel A) and in the text, considering that at 8 weeks the overall profile was not significantly different between the two diets. However, we prefer to maintain the time specific differences for the curves with a significant time · meal effect.
5. It is not clear to me that the difference in post prandial lipid patterns reflects the eight weeks prior to the meal test. The differences could just as easily represent acute differences in meal composition.
Response: As reported in the text, to untangle the impact of the acute meal effect and the 8-week effect of dietary intervention on postprandial lipid changes between groups, the differences observed after 8 weeks of dietary intervention were adjusted for the values of the baseline. The differences between the groups remained significant, with a greater effect in terms of percentage reduction particularly for triglycerides when comparing the multi-factorial diet with the MUFA diet. See lines 152-154 in the Statistical analysis paragraph, and lines 230-236 in the Results paragraph.
6. What evidence of compliance with dietary intervention do the authors have?
Response: as reported in the Material and Methods, compliance with the dietary treatments was evaluated by a 7-day food record filled in by the participants at 4 and 8 weeks, see lines 110-113.
The composition of the diets followed by the participants, average of the 7-day food records completed at weeks 4 and 8, strictly reflected the composition of the diets assigned to either group, with expected significant differences between groups in the amounts of the characterizing components as reported in detail (see reference 20, Della Pepa, G. et al. BMJ Open Diabetes Res Care 2020, 8, e001342).
Furthermore, as reported in the Discussion, direct evidence regarding optimal dietary adherence is confirmed by the variations in the serum triglyceride fatty acid profile of the participant following the two dietary interventions. In fact, in our previous paper (see reference 21, Costabile, G. et al. Nutrients 2022, 14, 2178), we have shown that plasma triglycerides were significantly richer in PUFAs after the multifactorial diet (higher in PUFAs), whereas a significant increase in oleic acid was observed after both experimental dietary interventions (both higher in oleic acid coming from the extra virgin olive oil, the major source of the MUFA characterizing the two diets).
We have clarified this point in the revised manuscript.
“…the optimal adherence to the two experimental diets, also confirmed by the variations in the serum triglyceride fatty acid profile ‒i.e., PUFAs significantly increased in the triglycerides only after the multifactorial diet, whereas a significant increase in oleic acid (the major source of the MUFA characterizing the two diets) was observed after both experimental dietary interventions”. See lines 319-323.
7. Lowering HDL with a multifactorial diet seems like an unintended adverse event.
Response: in our study, the multifactorial diet induced a greater decrease in HDL cholesterol compared to the MUFA diet, although this effect was not statistically significant. This finding might apparently be considered an adverse effect of the dietary intervention; however, we cannot exclude a possible effect on HDL functional properties beyond their concentrations, highlighting the paradigmatic relevance of HDL quality rather than quantity, in accordance with most recent evidence.
In the revised manuscript we have further expanded the discussion on this point.
“Previous studies had observed a reduction in fasting HDL cholesterol levels after consuming diets rich in polyphenols and dietary fiber [19, 37-40], and this finding, at first glance, could be considered an adverse effect in terms of CVD risk. However, evidence shows that the beneficial effects of HDL in decreasing CVD risk are not fully reflected through the amount of cholesterol carried by the HDL; rather, the quality of the HDL particles, i.e., their functions in cholesterol efflux capacity and their beneficial properties in oxidative stress, inflammation, and endothelial function [40], may be more relevant. In our study, we cannot exclude that, despite the observed reduction in HDL cholesterol, the multifactorial diet could have improved the quality composition and function of HDL cholesterol as reported with a similar dietary pattern [40,41]. The impact of different dietary components on HDL cholesterol concentrations and functional characteristics in the postprandial state and its role in the modulation of CVD risk is poorly understood, indicating a need for additional research.” See lines 305-317.
8. The authors might want to consider changing the name of “multifactorial diet” to something like “prudent diet” or some other title more descriptive than the title they’ve chosen.
Response: We thank the reviewer for this point. However, we prefer not to change the definition of “Multifactorial” considering that in our previous published papers (on the primary outcome of the study as well as on the ancillary analyses), the term “Multifactorial” has been reported. The change of term in the current paper might be confusing with regard to the previous papers. When we designed the clinical trial and elaborated the experimental diet, we chose to use the term “Multifactorial” to give particular emphasis to the different beneficial dietary components characterizing the dietary intervention beyond the possible health effects that we could have observed at the end of the trial. On the other hand, our "Multifactorial Diet" presents some characteristics of other healthy dietary patterns, including the "Prudent Diet," and in the revised paper, we have considered this point in the discussion.
“Considering the various features of the multifactorial diet ‒i.e., whole grain, dietary fiber, and polyphenols‒ inspired by the traditional Mediterranean diet and sharing components of other healthy dietary patterns such as the Nordic diet and the Prudent diet, it is likely that its efficacy on postprandial lipid reduction might depend on the synergic effect of the different dietary components”. See lines 263-264.

Round 2
Reviewer 2 Report
Comments and Suggestions for Authors
The authors have satisfactorily responded to reviewer concerns.
Author Response
We thank the reviewer